# A Qualitative Study Exploring the Lives and Caring Practices of Young Carers of Stroke Survivors

**DOI:** 10.3390/ijerph19073941

**Published:** 2022-03-25

**Authors:** Trudi M. Cameron, Marion F. Walker, Rebecca J. Fisher

**Affiliations:** School of Medicine, Mental Health and Clinical Neurosciences, Queen’s Medical Centre, University of Nottingham, Nottingham NG7 2UH, UK; marion.walker@nottingham.ac.uk (M.F.W.); rebecca.fisher@nottingham.ac.uk (R.J.F.)

**Keywords:** young carer, stroke, families, children, agency, information, qualitative

## Abstract

This study investigated the life transformation of children when a parent returned home from hospital after a stroke. The study explored children’s experiences of taking on caring roles in partnership with their well parent and stroke survivors’ perceptions of the young carer roles. The study aimed to identify unmet support needs in order to inform future service provision. Semi-structured interviews were conducted separately with stroke survivors (*n* = 7) (age range 41–60 years, mean 50.6) and their young carers (*n* = 11) (age range 11–20 years, mean 16). Interviews were audio recorded, transcribed and analysed using reflexive thematic analysis. Three overarching themes were identified: the lives of young carers, impact of stroke, and insulating the family. All the children were providing some level of care. None were in receipt of any formal support. Children moved in and out of the caring role in the absence of an adult, to act as primary carer. Young carers valued the life skills they were gaining but reported gaps in their knowledge and understanding of stroke. The well parent and young carers worked together in a bi-directional partnership to ameliorate the impact of stroke on the family. The study concluded that age appropriate and stroke specific information for children of stroke survivors could enhance agency and optimise positive outcomes of caring.

## 1. Introduction

Research focused on young carers has successfully influenced national policy and legislation in the U.K. [1,2,3,4,5,6], strengthening young carers’ rights to assessment and support. This has led to parallel development of interventions to support them. 

Much of the extant literature pertains to young caring in general, with some condition specific research for conditions such as for multiple sclerosis [7,8,9] and Parkinson’s disease [10,11]. No evidence base exists specifically for young carers of stroke survivors [12]. Stroke differs from many other conditions as it occurs suddenly when the blood supply to the brain is occluded or there is a bleed resulting in damage to surrounding brain tissue. It is responsible for a wider range of disabilities than any other condition [13]. Effects can include weakness or loss of movement in any or all limbs, affecting mobility and ability to perform activities of daily living. It can also affect vision, communication, speech and language, hearing, balance, personality and mood [14]. The degree and rate of recovery are unpredictable and can continue for many years. Stroke has been traditionally perceived as a condition of older age hence there is no evidence base regarding the impact stroke can have on children or young carers of stroke survivors. Consequently, there are currently limited resources to signpost children and young people to, in the United Kingdom (U.K). As the demographics of stroke shift to a younger age group [15], front line health care workers are increasingly being faced with stroke survivors who have co-resident children, and may have limited experience and knowledge of working with and signposting them to appropriate support.

Young carers provide care in or outside of the family home, in either the long or short term, for people with physical or mental illness, disability, or who are misusing drugs or alcohol. Those aged under 18 years, referred to as young carers, are legally children who are protected under specific legislation in the U.K. [2,3,4,5]. Young people aged 18–25, referred to as young adult carers, also have legal rights in the U.K. to assessments at transition between adult and child services and access to financial benefits [4,5]. Care provided commonly includes: physical, such as assisting with personal hygiene, feeding or mobility, nursing care and assistance with medication; emotional, providing company and listening to worries; or practical, helping with cooking, shopping, cleaning, etc. The amount of time spent caring varies from a few minutes to many hours a week [16,17]. This has recently been reported in a national U.K. survey at over 90 hours a week as self-reported by some young carers [18,19]. When the young carer and their families have unmet needs, caring may have an adverse impact on the child’s health and well-being and transitions into adulthood [20,21].

The scale of young caring in the United Kingdom (U.K.) is still disputed [22,23,24,25]. A recent estimate suggested that between 2 and 8% of children in advanced industrialised, capitalist societies are carers [22] (p. 751). It is estimated that 7% of children in the U.K. are providing care, and 3% are providing a high amount of care [26]. Variations in estimates are due to the hidden nature and the lack of an agreed definition and of young caring. Some families and young carers do not recognise their role as carer [27]. Yet others choose not to reveal their caring role for multiple reasons including fear of unwanted statutory intervention, a sense of shame or embarrassment, and not wanting to appear different to peers [28].

Caring can provide positive and rewarding experiences for some young people [29,30,31]. Van de Port et al. [31] reported on a three year longitudinal study of children of stroke survivors. Positive outcomes of caring tasks included feeling more needed, having more responsibility, feeling more mature and having more time with their parents. However, that is not always the case and young people with caring responsibilities can also experience detriment to their health, well-being, development, and future life chances [6,20,32]. Impact can be significant and lifelong [29,33,34,35,36]. This negative valence is amplified when there are unmet needs for the young carer or within the family [37]. 

Children and young adults take on the role of young carer for various reasons [38,39,40,41,42]. Becker and Becker [38] identify situational triggers for young carers to take on unpaid caring roles drawn from the young carer literature. These include co-residency, and a lack of availability or willingness of an adult to take on the caring role. Taking a different approach, Kain [42] (p. 32) describes three mechanisms by which young carers take on their role: *“Embracing the Challenge”*, in which they observe and are intrigued by adults caring and wish to take on the skills themselves; *“Sharing the Load”*, in which concern about the negative impact of caring on adult members of the family drives an incentive to remove some of the burden; and *“Being Assigned”*. Young carers are said to be assigned when they are socialised to the role from a very early age with the gradual incremental increase in responsibilities rendering them unable to remember a time they were not caring [43]. The young carer may also be allocated the role by another family member who is unable or refuses to take on the caring role [44,45]. 

Many of the Becker and Becker [38] propositions are supported in other studies including a systematic review of young carers of parents with a chronic illness [36]. However, there are also findings in other studies which propose other causal factors such as young carers taking on their roles to preserve the family unit [46], and further, that children want to put the family first and try to hold the family together whilst attempting to maintain everyday life [36] (p. 6). Adolescents have been found to subjugate their own needs to prioritise those of the person they care for, as the family’s needs assumed a greater priority than their own. This extended to a limitation of their own future plans for further education and leaving home [8]. Other researchers suggest that lack of availability of statutory support for the cared for person also contributes to the young carer taking on the role [29]. Kain’s [42] model suggests an element of choice for some young carers whereas the Becker model [45] could suggest that agency of the young carers is constrained by the social, relational and contextual circumstances within which they live. This limitation of agency is likely to be aggravated by the lack of information young carers are reported to have about their rights to assessment and support, and the rights to support for the person they care for [47]. 

Silence, hiddenness, and invisibility of young carers have been key concepts reported consistently in many young carer studies. Aldridge and Becker [43] refer in their studies to a self-imposed silence resulting from young carers not wanting anyone outside of the family to know about their caring role, being afraid of the consequences of telling anyone or just finding it difficult to talk about. Silence has also been reported as being imposed by the parent not allowing the child to discuss their role outside the family [48]. It is usually reported as conscious decision and action. Hounsell [28] and others [49] refer to similar findings but categorise the concept as hiddenness. Hiddenness appears to result from active or passive behaviours, by young carers, parents and professionals, and can also be a response to societal norms [28,47]. It is possible that hiddenness and silence are constructs created by mutual decision and interactions between parent and child, and that children have more power and are more capable of making their own informed decisions than they are sometimes credited with. Kuczynski et al. [50] suggest that children are far from being passive recipients of one-way discipline and control, and consequently become products and victims of parental actions which he terms *“a unilateral model of parent child interactions”* (p. 25). Instead, they suggest that parent and child interact together to develop and make changes rather than reacting to each other. They call this: *“a bilateral model of parent child relations”* (p. 27). The model proposes that children and parents have equal agency and that children are capable of filtering, adapting and judging the appropriateness of parental messages and wishes [51]. Further, it proposes that effectiveness of the child’s agency is dependent on their developmental stage, the responsiveness of parents and external support [52]. This study aimed to investigate the lived experience of young carers of stroke survivors, as little was known about their existence and needs due to the hidden nature of their caring role. 

This paper explores for the first time how children of stroke survivors experienced and embraced the role of carer for their stroke surviving parent in a bi-directional partnership with the well parent. We also examine the challenges they faced in doing so, in the early stages of stroke recovery. The ways in which the young carer’s agency could be enhanced are also considered. 

All children and young adult participants are referred to as young carers throughout this paper for consistency and ease. 

## 2. Materials and Methods

### 2.1. Design

A qualitative design using semi-structured interviews was employed. Stroke survivors and their young carers were invited to describe the impact of stroke on their lives. 

### 2.2. Participants

Participants were recruited based on a cross-sectional survey of stroke survivors who were living in the East Midlands [12]. All stroke survivors indicating an interest and meeting the study criteria (Table 1) were provided with participant information sheets and were contacted one week later to arrange the interview at a time and place convenient for them. Written informed consent was obtained at the beginning of the interview. The stroke survivor and at least one of their young carers had to agree to participate. Parental consent and young carer assent were obtained for young carers under the age of 16 years. 

### 2.3. Procedure of the Interviews

Three semi-structured interview guides were developed for young carers aged 11–18 years, young carers aged 19–24 years, and stroke survivors. Guides were developed taking in to account relevant literature as presented in the introduction, and in consultation with key stakeholders. These included a Young People’s Advisory Group (YPAG) (the YPAG consisted of five regionally situated groups comprising members aged between 8–19 years trained to critique and advise on the development and conduct of research studies), lay assessors (members of the public, with experience of using health care services and, trained to appraise and advise on research design and implementation [53]), adult carers of stroke survivors, two young carers, and two lead managers of young carer projects. The interview experience was discussed with the first set of interview participants and the order of questions changed as a result to improve the flow. New areas for discussion that emerged during interviews were added to the interview schedule and posed to future participants, a technique classified as responsive interviewing [54]. Areas explored during the interviews are shown in Table 2. 

### 2.4. Data Management and Transcription

Interviews were audio recorded. Recordings were transcribed verbatim by the first author (TC) and were redacted to remove any identifying markers. Pseudonyms have been assigned to participants to ensure anonymity. All data were sorted and stored using a combination of manual techniques and computer assisted software QSR NVivo (version 11) for Windows to assist analysis. 

### 2.5. Analysis Procedures

Reflexive thematic analysis was adopted, utilising both an inductive and deductive approach to the analysis. Thematic analysis was used to identify and analyse meaningful patterns within data and was useful for understanding common meanings and shared experiences [55,56] Conducting the interviews and transcribing the recordings familiarised the first author (TC) with the corpus of data. Each transcript was read in hard copy and notes made of emergent key words and phrases. Transcripts were uploaded to NVivo 11 (QSR International UK, Cheshire, UK). Initial codes were devised from a word frequency query run in NVivo, notes made on the transcripts and a priori codes taken from the interview schedules. Codes were applied to the transcripts in an iterative process of revising codes and then recoding with the new codes. Themes were refined through discssion with the third author (RF). 

## 3. Results

Seven stroke survivors and 11 of their young carers participated. Pseudonymised vignettes describing participants can be seen in Table 3.

Stroke survivors and young carers were interviewed separately on the same day at the stroke survivor’s home between July and October 2017. Stroke survivor interviews lasted a mean of 42 min (range 18–103 min) and young carer interviews lasted a mean of 31 min (range 15–44 min). The majority of stroke survivors were male (5) with a mean age of 50.6 years (range 41–60 years). All were living with a spouse or partner who they considered the primary carer. However, one young carer was primary carer during the week as their father worked away. The average age of the young carers was 16 (range 11–20 years) and there was a larger proportion of females (7) than males (4). None of the young carers had been formally identified as carers and none had received a formal assessment of their needs.

Three themes and eight sub-themes were identified and are presented in Table 4. 

### 3.1. The Lives of Young Carers

All of the young carers interviewed were performing caring tasks for the stroke survivor. Only one was acting as a primary carer during the week while the stroke survivor’s partner was working away. For the majority, the stroke survivor’s spouse or partner was providing the personal care whilst the young carers back filled the other chores and tasks that the primary carer no longer had time to do. Young carers moved in and out of the caring role, including delivering personal care depending upon the availability of an adult to perform the primary caring tasks. With the exception of two young carers with previous caring roles, the majority did not identify with the term young carer and defined their contribution to caring as an extension of their routine family life. Their perception of a ‘carer’ was the person providing personal and intimate care, whether that was the stroke survivor’s partner or a paid carer. They did not perceive emotional and practical support as falling within the remit of carer. 


*“It is more of the normal family thing for us. My mum is more of the carer. Me and my sister, we are here to help with him but …its normal routine for us.”*

*(Alfie 17)*


The majority of parents in this study did not recognise their child as a carer, even when they were providing personal care. 


*“Carers to me come in from a company but family members to me aren’t carers they are just family.”*

*(Sue 41)*


The reallocation of tasks between the well parent and the young carers appeared to be a joint and mutually negotiated decision between them, such as would be expected in a bi-directional relationship. 

#### 3.1.1. Tasks of Caring

Five categories of caring tasks were identified: practical tasks, emotional caring, rehabilitation, personal care, and supporting the well parent. Tasks relating to each category can be seen in Table 5. 

In some families, chores had been part of a daily routine for children, increasing in complexity commensurate with age and ability. For others, children had not completed any chores or tasks in the home prior to the stroke and, therefore, small tasks such as preparing their own breakfast was a significant change. Emotional support for the stroke survivor was provided by all of the young carers in the study and included trying to maintain positivity and focus on recovery, particularly through rehabilitation exercises. 


*“He gets quite upset about what he realises what he can’t do. We try to keep him focused on what he can do to keep him motivated because he can move his right leg a bit and we keep getting him to do that.”*

*(Molly 20)*


Some young carers also helped with physiotherapy exercises, putting splints on, mobilisation, and speech therapy, and had learned by watching the professionals. None had been taught what to do.


*“To help my dad I did his speech and language thing on his IPad…. but he said it’s too babyish ….I helped him with that and I have done a few more chores around the house now because mum has to help him with other things. So I’ve had to do more or less some of hers.”*

*(Jamie 11)*


Personal care was limited to three young carers. The remainder were protected from having to deliver any personal care by the primary caring adult in the family and the stroke survivor. 


*“I like them [young carers] helping me but not when it comes to food or toilet.”*

*(Dennis 58)*


Most young carers reported enjoying the tasks they had taken on with the exception of one young carer.


*“The worst job? Probably taking the commode out because it stinks.”*

*(Declan 12)*


#### 3.1.2. Support Needs

The greatest support need expressed was to be recognised by health care professionals (HCPs) and given information on stroke. Most young carers had a limited knowledge and understanding of stroke which had been gained in an informal, unstructured and ad-hoc way. The majority had learnt about stroke by talking to their parents, listening in to conversations by doctors and nurses on ward rounds and handovers, and observing the impact of stroke on the stroke survivor they were helping to care for. 


*“I have [over] heard bits of what the doctor said but I’ve not really understood it properly. I just know it’s something to do with the brain.”*

*(Laura 17)*


None of the families could recall any specific sources of information about stroke for children or young carers, and no one had been offered any information, which, in turn, led to a lack of awareness of support services that may be available to them. The lack of access to information was largely due to a deficiency in signposting by health care professionals. However, in some cases, it was due to a lack of opportunity as not everyone had access to the internet at home due to financial constraints, or rurality limiting internet connectivity. One young carer said her grandfather had some written information, but as she had severe dyslexia she felt overwhelmed by the volume of information he had been given, and had not been able to access it. 

Health care professionals were seen as being task focused, failing to recognise or optimise opportunities to engage with and educate families and young carers in particular. This was during inpatient stays:


*“Even the nurses, they are very sort of, job and move on. They don’t really, tell you anything. The majority of things I learned from them was from when they were reading from their handovers to each other at the end of the bed.”*

*(Anna 20)*


And in the community: 


*“they [health care professionals] just do their job and then they skedaddle.”*

*(Jamie 11)*


Several young carers suggested that information delineating potential consequences of stroke would be useful. One young carer said that the physical effects were obvious but the mood swings and psychological problems such as depression could not be seen. Families needed to be warned to be on the lookout for such things and advised about what they could do to help. Most wanted help to support the stroke survivor more effectively. One young carer suggested that the stroke survivor could be supported by health care staff to identify a list of things they were struggling with and needed help from the family to address, and some suggestions for how those needs could be met. Some felt the information could be a starting point to help the family talk to each other about what had happened.


*“It would have been useful to have some information as a piece of paper or as an online resource that gave me a list of all the kinds of effects a stroke could have and that might have helped me to help her more now. Because apart from her mobility issue I really didn’t know what I could do to help her out. So maybe if I had known about the things she was struggling with I could have helped her a bit more….if there was a big handbook with all the different things that have been reported and then you could read through and then ask her, do you need any help with this?”*

*(Jack 20)*


#### 3.1.3. Benefits and Detriment of Caring

Most young carers and parents were positive about skills learned, experiences gained and attributes that had been enhanced. Practical skills were positively viewed as preparation for adult life. Experiences gained were seen as potential material to enhance a curriculum vitae for future job applications and two of the young carers were considering entering a caring profession in adult life. Attributes that were enhanced included being more compassionate, patient, tolerant and more considerate of others. Parents reported an increased independence in the young carers.


*“Well before I wouldn’t usually have got dressed, I’d have come straight downstairs and my breakfast would have been ready and my mum would have told me to go back upstairs and get my clothes on but now I usually get up, out of bed, do my teeth, do my clothes, come downstairs and make my own breakfast and then get my stuff ready for school. Do everything I need to do and then go to school.”*

*(Jamie)*



*“Do you have to get your sister ready for school as well?”*

*(researcher)*



*“She does it herself. I started doing it and then she copied.”*

*(Jamie)*



*“Do you walk your sister to school?”*

*(researcher)*



*“Yes I do.”*

*(Jamie 11)*


Very few detriments of caring were reported in this study. One young carer sometimes resented having to care for a younger sibling especially as deadlines loomed for college course work. Negative impacts were mostly related to the impact of the stroke rather than the caring role, such as worry and concern for the well parent. 


*“Is there anything you have had to do since he came home that you haven’t liked doing?”*

*(researcher)*



*“Maybe looking after my brother a bit when I’ve got things to do. But then I would always put him first.”*

*(Anna 20)*


### 3.2. Impact of Stroke

Stroke affected the whole family, the way they lived and related to each other. Young carers had to learn to deal with changed behaviours and personalities of the stroke survivor, physical disability, lability of emotions and the loss of the parent’s ability to parent as effectively as they had before the stroke. 

#### 3.2.1. Changing Relationships

Relationships changed both within and outside the family unit:


*“Everybody’s life changed. And the way they all approached me changed. I thought that before they always treated me nicely but I never realised how nicely they could treat me until I had the strokes.”*

*(Dennis 58)*


Friends had stopped visiting and older young carers were saddened by this.


*“….not being able to go out with his friends. Not being able to socialise with them as much as he would. It’s horrible.”*

*(Alfie 17)*


Young carers were aware of their changed relationships with the stroke survivor:


*“What about looking after you? Can she look after you the same?”*

*(researcher)*



*“No not really. But I look after her. So it’s alright. I look after myself and I look after her yeah.”*

*(Declan 12)*



*“Instead of him being there for you, you have to be there more for him.”*

*(Molly 20)*


#### 3.2.2. The Lived Experience of the Biopsychosocial Effects of Stroke

Some of the stroke survivors had profound disabilities, including being unable to walk or transfer from bed to chair unaided, communication and cognitive problems, visual problems, and lability of emotions. One stroke survivor was being tube fed and required suction to clear his airways. Pain and noise sensitivity caused irritability for several and all were experiencing extreme fatigue. Changes in behaviour and personality were noted by the young carers and caused concern when they did not understand what was happening or how to help.


*“…sometimes she is just alright and sometimes she gets really angry and it’s just confusing. …. Sometimes she just starts to cry and when we ask her are you crying? She says no. But we know that she is crying. She doesn’t want to...”*

*(Declan 12)*


#### 3.2.3. Living Differently

Stroke necessitated physical alterations and adaptions to the home, in addition to changes to living arrangements and lifestyles. However, none of the young carers expressed any resentment and were keen to ensure that the needs of the stroke survivor were met as far as possible. None of the households had been formally assessed for changes to make the living space more adaptable or useable. This was attributed by participants to uncertainty over recovery trajectory and statutory agencies waiting to see how much recovery would be made. 

Some families had made their own adaptions including building ramps and converting downstairs rooms to toilets and bedrooms. 


*“Me and my brothers we built this (points to ramps) out the back so we could get the wheelchair inside. We built this ramp and concreted all around the sides. We’ve just done what we can to make it work more than anything. We put the canopy thing out there as well. So between us we got it so everywhere is liveable. I carried the bed down so.. we just had to get it done.”*

*(Ben 20)*


Others had devised aids such as book holders and an adaption to a fishing rod to enable stroke survivors to re-engage in favourite past times. Several of the stroke survivors were unable to manage stairs and were living, sleeping and toileting in the only shared lounge space. This meant that young carers were no longer able to have friends to visit their home. Young carers changed morning and evening routines to accommodate paid carers coming in, and to ensure they were not walking through the lounge during personal care, rest or bedtimes. 

Some young carers had stopped engaging in hobbies that they had done jointly with the stroke survivor. 


*“We used to do so much together, that was the biggest change, biggest shock really. We used shoot together on Tuesdays, if I had stuff to do on the car he would help me with the car, you know everything. We used to go fishing at the weekends, go out on the boat, all sorts of different things so yeah physically more than anything. You can talk to him and still have the same conversation with him. …..*
*I have stopped shooting now. We started when I was at school. Probably about six years maybe we were shooting together. But it didn’t feel right going without him. I didn’t really want to go. So we stopped doing that.”*

*(Ben 20)*


Some of the older young carers had reduced social time out with friends to be closer to home in case anything happened. For the one primary young carer in the study Declan (aged 12), life revolved around caring for his mother. He *“made do”* with using kerbs outside of the house instead of going to the local skate park, to make sure he was near if his mother needed him. He no longer had friends over to play as his mother’s bed was now in the living room. He would come straight home from school, do any necessary jobs and would go to his friend’s house but always with his mobile phone in case she needed him. He slept downstairs on the settee during the week while his father was away working in case his mother needed anything in the night.


*“I just want to stay here. To be here. I’d love to be home schooled so I could check on my mum and work at the same time yeah.”*

*(Declan 12)*


### 3.3. Insulating the Family

The importance of family and the desire to protect and support each other whilst dealing with all the uncertainties they were facing was clear. 

#### 3.3.1. Supporting Each Other

Young carers were watchful of the well parent and were mindful of the additional stresses placed upon them. They attempted to mitigate some of the additional load they could see had been generated by caring responsibilities and the tasks stroke survivors were no longer able to perform.


*“She [mother] does most things. We help her out when she needs some help. But I think she’s got a hard job if I am honest. He’s not easy to look after.”*

*(Alfie 17)*


This support was not time limited and several of the young carers were planning to remain living near or at home in the long term to help.


*“We [young carer and partner] are hoping to have somewhere close to here because I will want to have [sisters] around to give mum a break from them and obviously I want to see them as well. So I want somewhere they can walk to …and if they want to sleep over for a few nights. Somewhere they can walk to school.”*

*(Anna 20)*


#### 3.3.2. Protecting Each Other

Families invested a great deal of energy in protecting each other from negative feelings and worries by not talking about stroke. Young carers talked about keeping the stroke survivor company and watching films together, but not being able to talk about the stroke for fear of upsetting them.


*“We never really speak about the stroke though because it’s a hard subject to touch on, do you know what I mean? It’s hard for him, rather than us.”*

*(Alfie 17)*


Alfie’s stepfather attributed his withdrawal from social life to his fear of having another stroke in public. He spoke at length about his realisation of his own mortality, the risk of having another stroke, how distressed he was about the impact his stroke had wrought on the rest of the family and about being a burden on them. Conversely, his young carers were happy to provide as much support as he needed and shared the same fears.


*“Is that your fear that he is going to have another stroke?”*

*(researcher)*



*“Yeah definitely. And then I am worried about him worrying about having another one, so it doesn’t really stop.”*

*(Laura 17)*


Some stroke survivors were expending a great deal of energy in hiding their innermost feelings and some of the physical and cognitive challenges they were facing from the young carers. One stroke survivor described how he did not like his young carer to see him depressed, and so pretended to be upbeat when she was around.


*“Generally they [young carers] lift my spirits but occasionally I have to act.”*

*(Dennis 58)*


Another of the stroke survivors spoke about how she had concealed her stroke diagnosis initially from the children and the cognitive and emotional effects of her stroke. This was exhausting for her but appeared to be quite effective as the young carers had limited awareness of the problems she was facing and were positive that she had made a good recovery. 

Most of the young carers were extremely positive about their role. One young carer explained how they managed to provide all the care their stepfather required as they knew that he would be upset by a stranger coming in to the home to provide personal care. 


*“We’ve just done it together. We would rather do it as a family than him be with someone he is uncomfortable with and he doesn’t know.”*

*(Alfie 17)*


Others did not share information about their caring role outside of the family. 


*“It’s just something you try to keep in the family.”*

*(Molly 20)*


## 4. Discussion

To our knowledge this was the first qualitative study exploring how the lives of children were changed and impacted when a parent or grandparent returned home from hospital following a stroke. All of the young people in the study had taken on additional roles to contribute either directly or indirectly to the care of the affected parent. They moved in and out of the caring role dependent on availability of an adult to act as primary carer. Most did not identify themselves as a young carer and none had been identified as such, referred for an assessment or offered any information, signposting or support. Young carers reported a lack of awareness of their existence by HCPs and felt invisible [40,57].

The young carers in this study were very positive about their caring role at this early stage in the stroke pathway [40]. Young carers reported acquisition of beneficial skills and attributes obtained from the caring role such as developing maturity and responsibility, closer family relationships, feeling valued, learning life skills, gaining independence and increased self-esteem. Many other studies have replicated these findings [40,48,58]. However, this should be treated with caution as some studies have reported young carers feeling coerced into taking on the role and resentful of it [40]. It is possible that negative effects were limited in this study as the majority of study participants were not primary carer or providing full time personal care. This was also relatively early in the stroke pathway and, therefore, long term effects of caring were not assessed. 

Five categories of tasks were identified that young carers were involved in: emotional caring, physical tasks, personal care, helping with rehabilitation, and supporting the well parent. Four of the categories are in common with the wider young carer literature and recognised in assessment tools in common use in the U.K. [59]. However, helping with rehabilitation appeared to be unique to stroke. This was potentially due to the model of rehabilitation in place at the time of the study. This involved the same type and intensity of therapy being delivered in the home as stroke survivors would have received if they were still in hospital [60,61]. Young carers were, therefore, exposed to the concept of rehabilitation and often witnessed therapy sessions, as the study took place during the U.K. six week long summer holidays. Some young carers were intrigued and could identify that helping with the therapy in between sessions was something they could do to proactively help to move the affected parent back to full health. This resonates with Kain’s [42] mechanism of “embracing the challenge” (p 32) in becoming a young carer. However, none of them had been shown how to deliver therapy safely or appropriately, potentially putting themselves and the stroke survivor at risk of injury [62]. Previous studies have shown that family involvement in rehabilitation improves outcomes [63] and, therefore, increasing the competency and agency of young carers in delivering rehabilitation could potentially improve outcomes for stroke survivors. 

The findings from this study provide an opportunity to reconsider the way in which young carers are often viewed as victims of circumstances. Previous young carer studies, whilst not entirely focused on a unidirectional model, have not explicitly explored the potential for adopting a bi-directional view of parent–child relationships, although it is implicit in some studies. The unidirectional model has performed a valuable function when used by the media to capture the attention of the public by portraying young carers as victims of circumstances [40]. This has served an important function in raising awareness of the existence of young carers, acting as a catalyst to change legislation, policy and practice around young carers’ rights. However, Kuczynski et al. [50] suggest that some fields of child and family studies are constrained by the use of a unidirectional model. This could potentially limit opportunities to think innovatively about future interventions for young carers and their families. Innovation in the future will be key to managing the financial and resource constraints currently being experienced by statutory services in the U.K. 

This study showed how the young carers and parents were working together in partnership to deal with the after-effects of stroke that impacted every aspect of their lives. The study was unique as it sought out young carers who had not been identified by statutory agencies. As a result, they provided an unadulterated view of their lives, which they reported as being positive despite the huge changes in their lives. Studies have shown that families close ranks and do not always share the real pressures and difficulties they are facing [46], and, therefore, the positivity presented must be viewed with caution. Families supported and protected each other and at this early stage in the stroke pathway, this reciprocity seemed to be sufficient to allow them to meet their own and each other’s needs. It is not known whether this state of positivity will persist beyond the early phases of the stroke recovery. Viewing these findings with a bi-directional lens, it is possible to see that children and parents potentially have the resources between them to deal with many of the new challenges they face. However, the lack of awareness and engagement by front line health and social care staff meant that opportunities to optimise the agency of the young carers simply by provision of timely and appropriate information and signposting, was missed. By rectifying this omission, some of the unmet needs that have previously been shown to have potentially detrimental effect on the young person’s health, well-being and future life chances [48] could be addressed. Being fully informed about their parent’s condition has been reported to improve psychological outcomes for adolescents [64]. 

The sense of family and the need to protect each other was clearly evident, however, families did not know how to talk to each other about stroke, or their fears and uncertainties about the future. Mauseth and Hjalmhult [8] reported that adolescents of parents with multiple sclerosis experienced improved well-being where there was openness about the illness within the family and wider networks. Further, it was influenced by knowledge of the disease and support from healthcare professionals. They recommended long term support from health professionals offering information and guidance. Families need to be supported and enabled to start the conversation in order for support needs to be identified and addressed. Both of these approaches have recently successfully been tested in a European study of whole family approach to young carers [65]. Stroke specific information could be developed similar to that produced for other long term conditions with a well-established evidence base for supporting children and young carers of affected adults such as multiple sclerosis and Parkinson’s disease. This could also be used as a tool to encourage communication within the family. 

The current financial climate and pressures on the NHS, further exacerbated by the COVID-19 pandemic, is accelerating the need to reconsider and redesign the way in which health and social care are delivered. Support must be targeted at those most in need and a hierarchy of interventions could be considered. However, assumptions cannot be made about who those people are, and as some of the young carers reported, Health care professionals (HCPs) involved with these families appeared unaware of the contribution of the children in the family to the caring role. A tiered model of support, similar to the stepped model of psychological support [66], would allow families and young carers to exercise their agency and control if, and when, support was needed. This would need to be developed in consensus with stroke survivors, families and HCPs, but could range from signposting to reliable information sources and potential support, to full young carer assessment and referral to formal support. More resources should be focused on recognition and enablement of internal family resources to limit unmet needs, enhance outcomes and place less demand on over-stretched statutory services. 

### Strengths and Limitations

This was the first study in the U.K. of young carers of stroke survivors. The young carers were unknown to statutory services and consequently had not received any assessment or offer of formal support. They, therefore, reported their lived experience from their own unadulterated perspective, providing a unique insight into their lives, their understanding and experience of stroke and caring for a stroke survivor. 

Limitations included a small sample size and the population being specific to one region in the U.K. The study population could have been biased by self-selection as only families who thought they may have young carers, or were prepared to admit they had young carers, were included. A further limitation was that the interviews were conducted during the six week national summer school holidays, and so the full impact of caring with additional pressure of attending college or school may not have been reported. Future studies will need to focus on the specific needs of the different age groups. Under 11 years were not included at all, and the needs of an 11 years old will be very different to those making transitions to adulthood at 16 and above. 

## 5. Conclusions

This was the first exploration of the potential to utilise a bi-directional model of parent–child interaction within the field of young carer research, and has the potential to change the way young carers are viewed and supported. 

In this study, children of stroke survivors became young carers when a parent or grandparent returned home from hospital following a stroke. They stepped into the caring role to fill the gaps created by the absence of an adult to fulfil the caring role as and when required, or were back filling roles that the well parent could no longer fulfil due to them taking on the primary caring role. None of the young carers had been formally identified or assessed, and none were in receipt of formal support from statutory agencies. At this early stage in the stroke pathway, families were working together to fulfil each other’s needs. However, they reported gaps in knowledge of information about stroke and referral pathways to support, and they did not know how to talk to each other about what was happening. Previous research has identified that knowledge of a parental condition can improve outcomes for young carers and the cared for. Age appropriate and stroke specific information that is accurate and accessible at the required time could help to mitigate some of the negative impacts of caring in the face of unmet needs. Statutory services have an opportunity, in the wake of the implementation of national Life After Stroke programmes, to educate front line staff in the identification and support of children and young carers of stroke survivors who have unmet needs. This could be further enhanced by incorporating a family approach to these initiatives, as recommended by the Together Project [65] and Becker et al. [67]. Not all young carers are in need of support all the time, and the development of a tiered support system could target those in most need, conserving scarce resources. At the time of writing there is a review of life after stroke services being undertaken by the Stroke Association to strengthen and deliver the policy directives in the NHS Long Term Plan, NHS Outcomes Framework and standards in the NICE and Royal College of Physician stroke guidelines [61,68,69,70], providing a unique opportunity to address current gaps in health and social care professionals’ training and awareness, and identification of children and young carers of stroke survivors in need of support. 

## Figures and Tables

**Table 1 ijerph-19-03941-t001:** Inclusion and exclusion criteria for participants.

	Inclusion Criteria	Exclusion Criteria
Stroke survivors	Aged ≥ 18 years. Living at home in the East Midlands. Returned a survey and consented to be contacted for an interview. Receiving support from a family member or friend aged 11–24 years. Able to communicate verbally and in English. The family member or friend aged 11–24 must also consent to the study. Young people <16 years assent to take part and their parent/legal guardian also consent.	Unable to communicate verbally in English. Living in residential care. Living outside of the East Midlands. Young person supporting them does not consent to take part in the study. The young person aged <16 years of age, does not give assent to take part in the study and/or the parent or legal guardian do not consent.
Young carers	Young person aged 11–24 identified by the stroke survivor as providing them with unpaid care or support.	Unable to communicate verbally in English.
	The young person and the stroke survivor both consent/assent to take part in the study.	Stroke survivor does not consent to take part in the study.
	Young people <16 years assent to participate in the study and the parent\legal guardian also consents.	Unable to meet inclusion criteria.

**Table 2 ijerph-19-03941-t002:** Interview topics.

	Interview Topics
Stroke survivors	Impact and effects of their stroke.
	Perception of the role and impact of caring on their young carers.
	Support needs of young carers.
	Suggestions for future improvements of family support.
Young carers	Knowledge of stroke and how that was ascertained.
	Insights into the impact of the stroke on the stroke survivor.
	Involvement in the care of the stroke survivor.
	Awareness of changes to individual and family roles and dynamics.
	Perception of benefits and detriments of caring and supporting the stroke survivor.
	Friendships and other support, including educational and vocational support and other coping mechanisms.
	Future life and career plans.
	Ideas for improving services and experiences for future families of stroke survivors.

**Table 3 ijerph-19-03941-t003:** Vignettes of stroke survivor participants and their young carers.

Stroke Survivor	Young Carer(s)	Notes
Dennis, male, aged 58 years, 366 days post stroke and home 125 days.	Son, Ben (joiner), aged 20, and Ben’s partner, Molly (college student) aged 20.	Lives with wife (primary carer), son and his partner. Wife resigned from paid employment to care. Dennis was profoundly affected by a brain stem stroke. No swallow reflex, fed by a tube directly inserted into the stomach via the abdominal wall, wheelchair dependent. Conservatory converted by the family to a ground floor bedroom. Paid carers daily. Attending rehabilitation unit twice weekly for physio and a shower.
Geoff, male, aged 46 years, 228 days post stroke and home for 104 days.	Son, Jamie (school child), aged 11, and daughter, not meeting inclusion criteria.	Lives with wife (primary carer) and children aged 11 and 9. No paid carers. Attends hospital for rehabilitation. Home rehabilitation has ended. Profoundly affected by stroke. Wheelchair dependent and communication problems.
Kevin, male, aged 50 years, 78 days post stroke and home for 70 days.	Stepdaughters, Alfie, aged 17 (college student),	Lives with wife (primary carer), two stepdaughters, a 5 years old son and grandchild < 1 year. Oldest son not resident and declined to participate in the study. Wife resigned from paid employment to care. Problems include: visual, cognitive, concentration, communication, impaired balance and mobility, sensory issues with sensitivity to noise and touch and nerve pain.
	Sally, aged 13 (school child);	
	son declined to participate.	
Jim, male, aged 56 years, 44 days post stroke and home for 40 days.	Granddaughter Laura, aged 17 (college student).	Lives with wife (primary carer) and granddaughter. Co-morbidities include severe depression and Parkinsonism. Physio and occupational therapy at home. Hemiplegia and balance problems but walks unaided.
Les, male, aged 43 years, 140 days post stroke and home for 56 days.	Stepdaughters, Anna, aged 20 (sales assistant),	Lives with wife (primary carer), three stepdaughters and one of their boyfriends. Wife resigned from paid employment to care. Wheelchair dependent, communication and cognitive impairment, clinically depressed. The bed and commode were in the lounge, which was the only shared space for the family. Paid carers attending daily.
	Lisa, aged 15 (school child).	
Sue, female, aged 41 years, 124 days post stroke and home 14 days.	Sons, Declan, aged 12 (School child), and older brother, declined to participate.	Lives with partner (long distance lorry driver away during the week), two sons aged 17 and 12. The 12 years old provides most care during the week. The 17 years old, in full-time apprenticeship and also a primary carer, declined interview. Co-morbidities including rheumatoid arthritis and renal failure, two previous strokes. House is very isolated with no internet connection. Hospital bed and commode in the only downstairs room. Washes at kitchen sink. No paid carers due to remote location.
Jan, female, aged 60 years, 127 days post stroke and home for 127 days.	Daughter, Faye, aged 14 (school child), and son, Jack, aged 20 (university student).	Lives with husband and two children. Oldest son at University in term time. Husband has severe depression and both children are on autistic spectrum. Fully mobile and self-caring. Some continence problems and depression, very verbose and difficulties concentrating since stroke.

**Table 4 ijerph-19-03941-t004:** Themes and Sub-themes.

Themes	Sub-Themes
The lives of young carers.	Tasks of caring.
	Support needs.
	Benefits and detriment of caring.
Impact of stroke.	Changing relationships.
	The lived experience of the biopsychosocial effects of stroke.
	Living differently.
Insulating the family.	Supporting each other.
	Protecting each other.

**Table 5 ijerph-19-03941-t005:** Tasks of caring.

Category of Tasks	Examples of Tasks
Practical tasks	Shopping
	Cleaning
	Cooking
	Making drinks
	Washing and ironing clothes
	Walking the dog
	Pet care
	Cleaning floors
	Pushing the wheelchair
	Helping the stroke survivor to mobilise Fetching and carrying things
	Care of younger siblings and taking them to school
Emotional caring	Keeping stroke survivors company Listening to them
	Keeping spirits up
	Motivating them
Rehabilitation	Physiotherapy
	Putting splints on
	Speech therapy
Personal care	Washing
	Dressing
	Fetching and emptying commodes and urinal bottles
	Shaving
	Washing and cutting hair
Supporting the well parent	Backfilling household chores
	Listening to them
	Trying to be “good” and not complaining about cancelled plans, holidays, etc.

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
