# Peer review of "A Qualitative Study Exploring the Lives and Caring Practices of Young Carers of Stroke Survivors"

_ijerph, 2022, doi:10.3390/ijerph19073941_

Round 1

Reviewer 1 Report

Dear Authors
The manuscript presented to me for review presents very interesting issues concerning the support given to sick people by young adult family members. I have no objections to the substantive and methodological side of the work.
However, the bibliography requires correction - some items seem to be out of date (details in the appendix in the form of comments). I do not understand the selection of the study group - it is inconsistent with the definition quoted in table n 1. I am asking for possible clarification of this issue.
Yours faithfully, Reviewer

Author Response

Thank you for your helpful comments. Please find attached my responses

Reviewer comments

Author responses

The manuscript presented to me for review presents very interesting issues concerning the support given to sick people by young adult family members. I have no objections to the substantive and methodological side of the work.
However, the bibliography requires correction - some items seem to be out of date (details in the appendix in the form of comments). I do not understand the selection of the study group - it is inconsistent with the definition quoted in table n 1. I am asking for possible clarification of this issue.

I have clarified the definitions of young carers in the introduction and removed Table 1 with previous definition in. Young carers are children under the age of 18 years, young adult carers are aged 18-25.

The figures in the abstract show that our participants were actually aged between 11 and 20 years as no one over 20 years volunteered to participate.

The eligibility criteria was 11-25 years to capture anyone in the UK in the secondary school system and above.  Younger children were not included in this study as recommended in

The lives of young carers in England Omnibus survey report Appendices to research report January 2017 Sarah Cheesbrough, Carrie Harding, Hannah Webster and Luke Taylor - Kantar Public With Professor Jo Aldridge - Young Carers Research Group, Loughborough University

The following references are out of date

9. Arnaud SH. Some psychological characteristics of children of multiple sclerotics. Psychosomatic Medicine. 1959;21(1):8-22.

Replaced with

Bjorgvinsdottir K, Halldorsdottir S. Silent, invisible and unacknowledged: experiences of young caregivers of single parents diagnosed with multiple sclerosis. Scand J Caring Sci. 2014 Mar;28(1):38-48. doi: 10.1111/scs.12030. Epub 2013 Mar 28. PMID: 23550661

10. Schrag A, Morley D, Quinn N, Jahanshahi M. Impact of Parkinson's disease on patients' adolescent and adult children. 603

Parkinsonism & related disorders. 2004;10(7):391-7.

Removed

21. Aldridge J, Becker S. Children who care. Inside the world of young carers. Nottinghamshire: Department of Social Sciences, 623

Loughborough University; 1993.

Removed

25. van de Port IG, Visser-Meily AM, Post MW, Lindeman E. Long-term outcome in children of patients after stroke. Journal of 630

rehabilitation medicine. 2007;39(9):703-7.

This is the only study internationally regarding children of stroke survivors and needs to be included. There are no subsequent papers and this highlights the gap in the literature that this study is addressing 

Howard D. Number of child carers 'four times previous estimate': BBC; 2010 [Available from

This is an important reference as it highlights the controversy in the numbers of young carers stated in many research papers and young carer survey reports.

28. Aldridge J, Becker S. Children caring for Parents with Mental Illness: Perspectives of Young Carers, Parents and Professionals. 639

Bristol: The Policy Press; 2003

Removed

33. Becker S, Becker F. Service needs and delivery following the onset of caring amongst children and young adults: evidenced 648

based review. University of Nottingham; 2008

This is important for the structure of the study and is an original source

43. Frank J. Couldn't Care More: A study of young carers and their needs. London; 1995. 666

Deleted and replaced with Hounsell (2013)

44. Tatum C, Tucker S. The concealed consequences of caring: an examination of young carers in the community. Youth and Policy. 667

1998;61(Autumn):12-27. 668

Deleted and replaced with Hounsell (2013)

45. Aldridge J, Becker S. Punishing children for caring: The hidden cost of young carers. Children and Society. 1993;7(4):376-87. 669

Removed

46. Kuczynski L, Harach L, Bernadini SC. Psychology's child meets sociology's child. Agency, Influence and Power in parent-child 670

relationships. In: Berardo FM, Shehan CL, editors. Contemporary Perspectives on family research Through the eyes of the 671

child:revisioning children as active agents in family life. 1. Conneticut: JAI Press Inc.; 1999. 672

This is a seminal paper on bi-directional parenting and needs to be included

47. Bell RQ. Contributions of human infants to caregiving and social interaction. In: Lewis M, Rosenblum LA, editors. The effect 673

of the infant on its caregiver. Oxford: Wiley-Interscience; 1974. 674

This is a seminal paper

48. Kuczynski L. Beyond bidirectionality. Bilateral Conceptual Frameworks for Understanding Dynamics in Parent-Child 675

This is a seminal paper on bi-directional parenting and needs to be included

Relations. In: Kuczynski L, editor. Handbook of dynamics in parent-child relations. Thousand Oaks, California.: Sage; 2003. p. 1-24

This is a seminal paper on bi-directional parenting and needs to be included

51. Moore T, McArthur M. We're all in it together: supporting young carers and their families in Australia. Health & social care in 680

the community. 2007;15(6):561-8. 681

Replaced with Moore 2009

52. Armistead L, Klein K, Forehand R. Parental physical illness and child functioning. Clinical Psychology Review. 1995;15(5):409-682

22. 683

Replaced

53. Bolas H, Van Wersch A, Flynn D. The well-being of young people who care for a dependent relative: an interpretative 684

phenomenological analysis. Psychology & health. 2007;22(7):829-50. 685

Replaced

54. Lackey NR, Gates MF. Combining the analyses of three qualitative data sets in studying young caregivers. Journal of advanced 686

nursing. 1997;26(4):664-71. 687

Replaced

55. Frank J, Tatum C, Tucker S. On small shoulders: Learning from the experiences of former young carers. London: The Children's 688

Society; 1999.

Deleted

56. Joseph S, Becker S, Becker F, Regel S. Assessment of caring and its effects in young people: development of the 690

Multidimensional Assessment of Caring Activities Checklist (MACA-YC18) and the Positive and Negative Outcomes of Caring 691

Questionnaire (PANOC-YC20) for young carers. Child Care Health and Development. 2009;35(4):510-20

Updated with the 2012 paper

59. Bibby A, Becker S. Young Carers In Their Own Words. London: Calouste Gulbenkian Foundation; 2000

This needs to stay

Reviewer 2 Report

General comments:

Thank you for giving me the opportunity to review this manuscript. The manuscript addresses an important topic and holds interest not only for health care professionals but also social services providers and policy makers. The presented study is evidently well researched and touches on many facets of young carers’ experiences and the implications this role has for their lives. Below, I make detailed comments and suggestions, which I hope will be considered constructive and helpful in further strengthening the manuscript for publication.

Abstract:

  • It is not clear if dyads were interviewed jointly or separately. Did survivors report on their perceptions of the young carer’s roles and practices? Also, 7 survivors and 11 carers suggest more participants than in contained in the term ‘dyads’ in some instances. Perhaps ‘group interviews’ is a better term, if interviewed jointly? Please clarify these points.
  • Theme titled ‘the lives of young carers’ is quite vague. Can this be further specified? Reading what the theme means below makes me think that perhaps this more about the caring practices and duties rather than the lives of the young carers. Presumably, their lives are not reducible to caring?
  • Also not clear, if in the phrase ‘parents and young carers’ ‘parent’ refers to the stroke survivor or the ‘well’ parent. Clarify please.
  • Conclusion that children should be supported with information to be better at caring. What about going beyond information, for example expediting necessary adjustments in the home environment or extending systematic support with care for stroke survivors by trained and qualified health workers in the home, and respite to relieve the carer burden on parents and children (e.g., to avoid longer term carer burnout)? Given that the manuscript points to many potential adverse outcomes for young carers – both immediate (see lines 59-61) and over the life course (e.g., Table 1, lines 66-68, 92-93) - this seems to be an important point and the discussion of findings also makes more specific suggestions.

Introduction:

  • Before presenting the prevalence of ‘young caring’ (lines 53-56) it would be good to define this population for the reader, for example in terms of age range, caring duties performed and/or time dedicated to such duties. Referring the reader to a table for a definition is not sufficient.
  • I would not read Becker and Becker as presenting young carers as passive victims of circumstance. However, they highlight the social, relational, and contextual constraints within which young carers exert agency.
  • Conceptually, I find it potentially problematic that children and young adult participants are combined in one ‘young carer’ category. What is the age range in the present dataset? Presumably, children and young adult carers are at different developmental stages, with different short-term aspirations and priorities. They also have different capacities to inflect the nature of the relationship with the person they care for – who remains their parent and possibly an authoritative figure. Are the authors able to comment and differentiate in their findings along those lines?

Materials and Methods:

  • Line 135: What is the research question that is referred to here?
  • Line 136: If it’s more than 2, it is no longer a dyad.
  • I am missing information about when interviews were conducted (e.g., what year and over how many weeks/months), and if they were done in person, over the phone, via videolink, etc.? Did the settings give participants enough privacy not to be overheard by family members?
  • Lines 165-167: ‘Audio recordings were uploaded to password protected secure university server and recording devices were wiped immediately after the interview.’ This detailed information is not needed.
  • Add reference to the paragraph on analytic procedure. The abstract states a reflexive thematic analysis was conducted (Braun & Clarke?). Please specify here.

Results:

  • Lines 183-191: Minutes for interview duration can be rounded. Also, I understand that in some journals the demographic and procedural information given here is seen as results. However, please consider moving to Methods as sample description.
  • Section 3.1. could end with a sentence after the quote situating this finding vis-à-vis the broader literature on informal caring. I feel that this links in nicely with what we know about female informal care, and the present study shows how this perception extends to young carers regardless of their gender.
  • Could the results be further elevated by referring back to the bi-directional conceptualisation of the caring relationship set up in the front end of the manuscript?
  • There also seems to be material presented from adult primary carers (spouses). Can it be made clearer in methods and abstract who the study participants were?
  • Lines 419-421: “So you are all worried about each other and worried about upsetting each other so you don’t talk about it?” (Researcher) “No”. (Laura 17) - I don’t feel this part adds much, given that this is just repeating a researcher statement.

Spelling, grammar, and errors of expression:

  • Lines 32-33: edit for clarity
  • Lines 48, 51: word repetition, also ‘signposting’ is an awkward phrase (also elsewhere in the manuscript)
  • Line 86: ‘young carers of chronic illness’ is an awkward phrase
  • Lines 88-90: incomplete sentence
  • Line 321: ‘older young carers’ is an awkward phrase

The sentence structure is sometimes wordy. Make statements as succinct as possible. See examples below and look for other instances to tighten the language:

  • Line 64: delete ‘they took on’
  • Line 72: shorten sentence “The reasons that children and young adults take on the role of young carer are ascribed to various causes” for example to “children and young adults take on the role of young carer for various reasons”

Author Response

Thank you for your helpful comments. Please find my responses below.

Reviewer comments

Author responses

General comments:

Thank you for giving me the opportunity to review this manuscript. The manuscript addresses an important topic and holds interest not only for health care professionals but also social services providers and policy makers. The presented study is evidently well researched and touches on many facets of young carers’ experiences and the implications this role has for their lives. Below, I make detailed comments and suggestions, which I hope will be considered constructive and helpful in further strengthening the manuscript for publication.

Thank you

Abstract:

·         It is not clear if dyads were interviewed jointly or separately. Did survivors report on their perceptions of the young carer’s roles and practices? Also, 7 survivors and 11 carers suggest more participants than in contained in the term ‘dyads’ in some instances. Perhaps ‘group interviews’ is a better term, if interviewed jointly? Please clarify these points.

·         I have removed the word dyad. Interviews were conducted separately as mentioned in the text.

·          

Semi-structured interviews were conducted separately with of stroke survivors (n=7) (age range 41-60yrs mean 50.6) and their young carers (n=11) (age range 11-20yrs mean 16)

The study explored children’s experiences of taking on caring roles in partnership with their well parent and stroke survivors’ perceptions of the young carer roles

·         Theme titled ‘the lives of young carers’ is quite vague. Can this be further specified? Reading what the theme means below makes me think that perhaps this more about the caring practices and duties rather than the lives of the young carers. Presumably, their lives are not reducible to caring?

·         The study revealed that stroke had much wider impact on the lived experience of the young carers than just the caring role. As explained in the findings, it impacted on living space, the way they could interact with friends i.e. not bringing them home because the stroke survivor was living in the lounge, not going out in case a secondary stroke occurred etc. 

“A qualitative study exploring the lives and caring practices of young carers of stroke survivors”

·          

·         Also not clear, if in the phrase ‘parents and young carers’ ‘parent’ refers to the stroke survivor or the ‘well’ parent. Clarify please.

·         Text changed-

·         The well parent and young carers worked together

·         Conclusion that children should be supported with information to be better at caring. What about going beyond information, for example expediting necessary adjustments in the home environment or extending systematic support with care for stroke survivors by trained and qualified health workers in the home, and respite to relieve the carer burden on parents and children (e.g., to avoid longer term carer burnout)? Given that the manuscript points to many potential adverse outcomes for young carers – both immediate (see lines 59-61) and over the life course (e.g., Table 1, lines 66-68, 92-93) - this seems to be an important point and the discussion of findings also makes more specific suggestions.

·         Incorporated the recommendation of a family centred approach in the conclusion. I   don’t want to go any deeper as I want to conclusions to specifically relate to this study rather than something wider which would duplicate studies already published

“Statutory services have an opportunity in the wake of the implementation of national life after stroke programmes, to educate front line staff in the identification and support of children and young carers of stroke survivors who have unmet needs. This could be enhanced further by incorporating a family approach to these initiatives as recommended by the Together Project (65) and Becker (66)”.

Introduction:

·         Before presenting the prevalence of ‘young caring’ (lines 53-56) it would be good to define this population for the reader, for example in terms of age range, caring duties performed and/or time dedicated to such duties. Referring the reader to a table for a definition is not sufficient.

·         Table one removed.

PLACED AT LINE 55

Young carers provide care in or outside of the family home, either long or short term for people with physical or mental illness, disability or who are misusing drug or alcohol.   Those aged under 18 years, referred to as young carers, are legally children, and have legal rights and are protected under specific legislation in the United Kingdom (UK)(3-6). Young people aged 18-25, referred to as young adult carers, also have legal rights in the UK to assessments at transition between adult and child services and access to financial benefits(5, 6). Care provided may be: physical, such as assisting with personal hygiene, feeding or mobility; emotional, providing company and listening to worries; or practical, helping with cooking, shopping, cleaning etc. The amount of time spent caring varies from a few minutes to many hours a week(17). This has recently been reported in a national UK survey as increasing(18). When the young carer and their families have unmet needs, caring may have an adverse impact on the child’s health and well-being and transitions into adulthood (19, 20) 

·          

·         I would not read Becker and Becker as presenting young carers as passive victims of circumstance. However, they highlight the social, relational, and contextual constraints within which young carers exert agency.

·         the Becker model (44) could suggest a agency of the young carers is constrained by the social, relational and contextual circumstances within which they live. This limitation of agency is likely to be aggravated by the lack of information young carers are reported to have about their rights to assessment and support and the rights to support for the person they care for (46).

·         Conceptually, I find it potentially problematic that children and young adult participants are combined in one ‘young carer’ category. What is the age range in the present dataset? Presumably, children and young adult carers are at different developmental stages, with different short-term aspirations and priorities. They also have different capacities to inflect the nature of the relationship with the person they care for – who remains their parent and possibly an authoritative figure. Are the authors able to comment and differentiate in their findings along those lines?

·         I agree with your comments however this was the very first exploration of the lives of children after their parents experienced a stroke and as such it was designed to capture an overall flavour of the issues to establish where the deeper dives should be targeted. We are planning a study to look at the specific needs of different age groups taking in to account the areas we have identified as being specifically related to stroke. I am therefore delighted that you have raised this point as it validates our decisions regarding the future direction of the work in this field. I will acknowledge this gap in the strengths and limitations  

Materials and Methods:

·         Line 135: What is the research question that is referred to here?

A qualitative design using semi-structured interviews was employed. Stroke survivor and their young carers were invited to describe the impact of stroke on their lives

·         Line 136: If it’s more than 2, it is no longer a dyad.

·         Removed all dyads and replaced with “stroke survivors and their young carers”

·         I am missing information about when interviews were conducted (e.g., what year and over how many weeks/months), and if they were done in person, over the phone, via videolink, etc.? Did the settings give participants enough privacy not to be overheard by family members?

Stroke survivors and young carers were interviewed separately on the same day at the stroke survivor’s home between July-October 2017. 

The interviews were conducted in the patients home as stated at line 203 and therefore face to face interviews are implicit.

Interviews were conducted separately and therefore as much privacy as was possible was afforded each participant.

·         Lines 165-167: ‘Audio recordings were uploaded to password protected secure university server and recording devices were wiped immediately after the interview.’ This detailed information is not needed.

·         Removed

Interviews were audio recorded. Recordings were transcribed verbatim by the first author (TC) and were redacted to remove any identifying markers. Pseudonyms have been assigned to participants to ensure anonymity. All data were sorted and stored using a combination of manual techniques and computer assisted software QSR NVivo (version 11) for Windows to assist analysis.

·         Add reference to the paragraph on analytic procedure. The abstract states a reflexive thematic analysis was conducted (Braun & Clarke?). Please specify here.

·         done

Results:

·         Lines 183-191: Minutes for interview duration can be rounded. Also, I understand that in some journals the demographic and procedural information given here is seen as results. However, please consider moving to Methods as sample description.

·         Minutes have been rounded.

·         We prefer to leave these are results

·         Section 3.1. could end with a sentence after the quote situating this finding vis-à-vis the broader literature on informal caring. I feel that this links in nicely with what we know about female informal care, and the present study shows how this perception extends to young carers regardless of their gender.

·         I would not consider this relevant in the findings section

·         Could the results be further elevated by referring back to the bi-directional conceptualisation of the caring relationship set up in the front end of the manuscript?

Added to section 3.1

The reallocation of tasks between the well parent and the young carers appeared to be a joint and mutually negotiated decision between them such as would be expected in a bi-directional relationship.

·          

·         There also seems to be material presented from adult primary carers (spouses). Can it be made clearer in methods and abstract who the study participants were?

·         Two of the stroke survivors invited their spouses to stay for the interviews as they had difficulty with word finding in some circumstances. I have removed the quotes from the spouses.

·         I have replaced 3.1.3 one with

“Well before I wouldn’t usually have got dressed, I’d have come straight downstairs and my breakfast would have been ready and my mum would have told me to go back upstairs and get my clothes on but now I usually get up, out of bed, do my teeth, do my clothes, come downstairs and make my own breakfast and then get my stuff ready for school. Do everything I need to do and then go to school”. (Jamie)

“Do you have to get your sister ready for school as well? (researcher)

“She does it herself. I started doing it and then she copied”. (Jamie)

“Do you walk your sister to school?” (researcher)

“Yes I do” (Jamie 11).

3.2.3 spouse quote deleted

·          

·         Lines 419-421: “So you are all worried about each other and worried about upsetting each other so you don’t talk about it?” (Researcher) “No”. (Laura 17) - I don’t feel this part adds much, given that this is just repeating a researcher statement.

·         deleted

Spelling, grammar, and errors of expression:

·         Lines 32-33: edit for clarity

Research focused on young carers, has successfully influenced national policy and legislation in the UK (1-6), strengthening young carers’ rights to assessment and support. This has led to parallel development of interventions to support them.

·         Lines 48, 51: word repetition, also ‘signposting’ is an awkward phrase (also elsewhere in the manuscript)

·         I can’t see the repetition?

·         Sign posting is common terminology in health services and is well understood. I can’t think of a suitable succinct alternative

·         Line 86: ‘young carers of chronic illness’ is an awkward phrase

·         Changed to -

·         Young carers of parents with a chronic illness

·         Lines 88-90: incomplete sentence

·         I can’t see where this is?

·         Line 321: ‘older young carers’ is an awkward phrase

·         Removed older

The sentence structure is sometimes wordy. Make statements as succinct as possible. See examples below and look for other instances to tighten the language:

Done

·         Line 64: delete ‘they took on’

·         deleted

·         Line 72: shorten sentence “The reasons that children and young adults take on the role of young carer are ascribed to various causes” for example to “children and young adults take on the role of young carer for various reasons”

·         Done

Reviewer 3 Report

Thank you for giving me the opportunity to review this manuscript. This manuscript is one of the best qualitative studies that I've reviewed.

I have a few comments for the authors to consider:

1) Table 3: please change the table's title "interview themes" to avoid confusion with the overarching and sub-themes. 

2) In the methods section, please add a line or two to inform the readers that you did thematic or content analysis (I assume), and how it fits with your research questions.

3) Table 4, are the names of participants their real names? I am not sure if I missed this in the methods section, but please make sure that you are not using their real names and to mention that you used different names in the methods section.

4) Please add strengths and limitations to your study.

5) In table 2, does the "Unable to communicate verbally in English" exclusion apply to young carers. If yes, please add it to the young carer's exclusion criteria.

6) Please consider revising the title to indicate that this is a qualitative study. You may also need to consider adding "qualitative" as one of your keywords.

7) in the methods section (lines 150-151) "Guides were developed following a review of the literature, consultation with key stakeholders." 

Please cite the literature that you guided you to develop the interview guides. Also, I think this sentence is missing "and" prior to consultation.

Author Response

Thank you for the very helpful comments. Please see my responses in the table below.

Reviewer comments

Author responses

1) Table 3: please change the table's title "interview themes" to avoid confusion with the overarching and sub-themes. 

Changed to interview topics

2) In the methods section, please add a line or two to inform the readers that you did thematic or content analysis (I assume), and how it fits with your research questions.

Thematic analysis was adopted utilizing both an inductive and deductive approach to the analysis. Thematic analysis is used to identify and analyse meaningful patterns within data and is useful for understanding common meanings and shared experiences  

3) Table 4, are the names of participants their real names? I am not sure if I missed this in the methods section, but please make sure that you are not using their real names and to mention that you used different names in the methods section.

Pseudonyms have been assigned to participants to ensure anonymity.

4) Please add strengths and limitations to your study.

Strengths and limitations

This was the first study in the UK of young carers of stroke survivors. The young carers were unknown to statutory services and consequently had not received any assessment or offer of formal support. They therefore reported their lived experience from their own unadulterated perspective, providing a unique insight into their lives, their understanding and experience of stroke and caring for a stroke survivor.

Limitations included a small sample size and the population being specific to one region in the UK. The study population could have been biased by self-selection as only families who thought they may have young carers, or were prepared to admit they had young carers were included. A further limitation was that the interviews were conducted during the six week long national summer school holidays and so the full impact of caring with additional pressure of attending college or school may not have been experienced by some of the young carers.  

5) In table 2, does the "Unable to communicate verbally in English" exclusion apply to young carers. If yes, please add it to the young carer's exclusion criteria.

Added

6) Please consider revising the title to indicate that this is a qualitative study. You may also need to consider adding "qualitative" as one of your keywords.

Title revised following your  and another reviewer’s feedback - A qualitative study exploring the lives and caring practices of young carers of stroke survivors

Qualitative added as a key word

7) in the methods section (lines 150-151) "Guides were developed following a review of the literature, consultation with key stakeholders." Please cite the literature that you guided you to develop the interview guides. Also, I think this sentence is missing "and" prior to consultation.

Guides were developed taking in to account relevant literature as presented in the introduction, and in consultation with key stakeholders